# Big Batch Bayesian Active Learning by Considering Predictive Probabilities

**Sebastian W. Ober**
Biologics Engineering
AstraZeneca
Gaithersburg, MD, USA
sebastian.w.ober@gmail.com

**Samuel Power**
School of Mathematics
University of Bristol
Bristol, UK
sam.power@bristol.ac.uk

**Tom Diethe**
Centre for AI, Biopharmaceuticals R&D
AstraZeneca
Cambridge, UK
tom.diethe@astrazeneca.com

**Henry B. Moss**
School of Mathematics
Lancaster University
Lancaster, UK
h.moss@damtp.cam.ac.uk

## Abstract

We observe that BatchBALD, a popular acquisition function for batch Bayesian active learning for classification, can conflate epistemic and aleatoric uncertainty, leading to suboptimal performance. Motivated by this observation, we propose to focus on the predictive probabilities, which only exhibit epistemic uncertainty. The result is an acquisition function that not only performs better, but is also faster to evaluate, allowing for larger batches than before.

## 1 Introduction

Batch active learning attempts to acquire batches of data that will be the most informative in building a model. In the Bayesian active learning setting, where the surrogate model directly gives its certainty in the latent function, BatchBALD [7] has been a popular algorithm for batch acquisition in classification. By extending the work of Houlsby et al. [5] and considering the mutual information over the whole batch, BatchBALD increases batch diversity over a naïve greedy approach. However, BatchBALD has a number of issues. First, in its closed form, it is limited by its exponential memory cost in batch size as well as an exponential computation cost. Moreover, as we depict in Fig. 1, despite its use of the mutual information on the whole batch, it is still susceptible to choosing similar points. We argue that the latter of these is caused by a conflation of the surrogate model's epistemic uncertainty (reducible uncertainty in the parameters or function values) and aleatoric uncertainty (irreducible uncertainty due to noise),[1] motivating us to explore methods of focusing on the model's epistemic uncertainty alone. Coincidentally, we show that by focusing on the continuous space of predictive probabilities, we can avoid the excessive combinatorial cost of enumerating all possible discrete outputs.

## 2 Background

In its most general form, the goal of active learning is to train a model as data-efficiently as possible by acquiring the most informative set of new points. In this work, we focus on classification

---

[1]See e.g., Kendall and Gal [6] for more in-depth discussion of the role of epistemic and aleatoric uncertainty in Bayesian machine learning.

Workshop on Bayesian Decision-making and Uncertainty, 38th Conference on Neural Information Processing Systems (NeurIPS 2024).

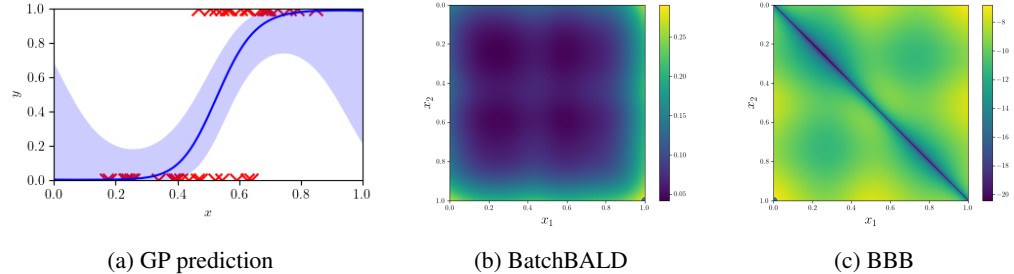

|  (a) GP prediction | (b) BatchBALD | (c) BBB |

Figure 1: The GP prediction along with BatchBALD and BBB-AL (ours) acquisition landscapes for $B = 2$, where $x_1$ (the first point in the batch) is on the x-axis and $x_2$ (the second point) is on the y-axis. For BatchBALD, we see that the optimal acquisition is $x_1 = x_2 = 1$, whereas for BBB-AL we obtain $x_1 = 0$, $x_2 = 1$.

tasks. We assume that we have already collected an initial dataset $\mathcal{D} = (\mathbf{X}, \mathbf{y})$ of input-output pairs, $\{\mathbf{x}_n, y_n\}_{n=1}^{N}$ with $\mathbf{x}_n \in \mathcal{X}$ and $y_n \in \{0, \ldots, C-1\}$, where $C$ is the number of classes. In batch active learning, we acquire a new batch of data $\mathcal{D}_B = \{\mathbf{x}_b, y_b\}_{b=1}^{B}$ by using a surrogate model trained on $\mathcal{D}$ to select a new batch of query points $\{\mathbf{x}_b\}_{b=1}^{B}$, which we send to an oracle to obtain $\{y_b\}_{b=1}^{B}$. We repeat this process multiple times, until we exhaust our budget of either data or computational resources. By using knowledge of previously acquired data, we hope that we can be more data-efficient than by simply acquiring a single large dataset from the start.

## 2.1 Bayesian active learning

In Bayesian active learning, we employ a Bayesian surrogate model, which imposes a prior distribution over functions, $\mathrm{p}(f)$, either explicitly, as with Gaussian process (GP) models [16], or implicitly through a prior distribution over model weights, $\mathrm{p}(\mathbf{w})$. Given a likelihood $\mathrm{p}(y|f, \mathbf{x})$ and data $\mathcal{D}$, we can obtain the posterior $\mathrm{p}(f|\mathcal{D})$ through Bayes' rule. This posterior is then used in tandem with an acquisition function to acquire the new batch of query points:

$$\{\mathbf{x}_1^*, \ldots, \mathbf{x}_B^*\} = \underset{\{\mathbf{x}_1, \ldots, \mathbf{x}_B\}}{\arg\max} \; a\left(\{\mathbf{x}_1, \ldots, \mathbf{x}_B\}; \mathrm{p}(f|\mathcal{D})\right).$$

These query points are passed to an oracle to obtain $\mathcal{D}_B$, and we update the dataset $\mathcal{D} \leftarrow \mathcal{D} \cup \mathcal{D}_B$.

For $B = 1$, a popular acquisition function for Bayesian active learning is known as Bayesian active learning by disagreement [BALD; 5], which takes the form

$$\begin{aligned} a_{BALD}\left(\mathbf{x}; \mathrm{p}(f|\mathcal{D})\right) &= \mathbb{I}\left(y; f|\mathbf{x}, \mathcal{D}\right) \\ &= \mathbb{H}\left(y|\mathbf{x}, \mathcal{D}\right) - \mathbb{E}_{\mathrm{p}(f|\mathcal{D})}\left[\mathbb{H}\left(y|\mathbf{x}, f, \mathcal{D}\right)\right], \end{aligned}$$

where $\mathbb{I}$ represents mutual information, and $\mathbb{H}$ represents differential entropy. Intuitively, this acquisition function favors points where the model is uncertain in its prediction (the first term) due to diverging hypotheses about the predictions (the second term, which penalizes agreeing hypotheses), i.e., where it has high epistemic uncertainty about the label of the point.

## 2.2 The promises and pitfalls of BatchBALD

A naïve extension of BALD to the batch case would be to simply take the points with the top-$B$ BALD scores. However, this will lead to redundant points; instead, Kirsch et al. [7] extend BALD to account for batches by considering the mutual information of the whole batch and the model's predictions:

$$\begin{aligned} a_{BatchBALD}\left(\{\mathbf{x}_b\}_{b=1}^{B}; \mathrm{p}(f|\mathcal{D})\right) &= \mathbb{I}\left(\{y_b\}_{b=1}^{B}; f \middle| \{\mathbf{x}\}_{b=1}^{B}, \mathcal{D}\right) \\ &= \mathbb{H}\left(\{y_b\}_{b=1}^{B} \middle| \{\mathbf{x}_b\}_{b=1}^{B}, \mathcal{D}\right) \\ &\quad - \mathbb{E}_{\mathrm{p}(f|\mathcal{D})}\left[\mathbb{H}\left(\{y_b\}_{b=1}^{B} \middle| \{\mathbf{x}_b\}_{b=1}^{B}, f, \mathcal{D}\right)\right]. \end{aligned}$$

While this appears to be a trivial change, it is not practical to implement in its naïve form. Therefore, Kirsch et al. [7] propose a couple of modifications: 1) a greedy approximation algorithm to avoid a combinatorial explosion in the joint scoring of sets of points and 2) Monte Carlo sampling to reduce the computational and memory costs for larger batches.

Despite its reliance on the joint mutual information considering the entire batch, BatchBALD does not fully mitigate the issue of querying similar points. Consider the simple binary classification problem shown in Fig. 1. We show the GP prediction for toy data in Fig. 1a, as well as the acquisition landscape for $B = 2$ for BatchBALD in Fig. 1b. We observe that there is not a significant penalty for acquiring twice in the same location, leading BatchBALD to select $x_1 = x_2 = 1$. This effect is caused by the inability of BatchBALD to distinguish effectively between epistemic and aleatoric uncertainty, meaning that the increased aleatoric uncertainty in the vicinity of $x = 1$ adds enough uncertainty to reduce the impact that considering the joint entropy of the batch has. By contrast, our method, BBB-AL (big batch Bayesian active learning), shows the desirable behavior of choosing distinct points, as shown in Fig. 1c, and directly penalizing the acquisition of similar points, as seen by the sharp diagonal area with low score.

## 3 Active learning for classification through predictive probabilities

Motivated by these shortcomings of BatchBALD, in this work, instead of focusing on the mutual information between labels and functions (or parameters), we propose to instead reduce the uncertainty in the predictive probabilities of the model. Typically, in classification, the surrogate model will model latent functions for each class, $f : \mathcal{X} \rightarrow \mathbb{R}^C$, where the class probabilities are given by a link function $\sigma(\cdot)$ which "squashes" the latent functions to $[0, 1]^C$ (e.g., a softmax): $\mathbf{p}(\mathbf{x}) := (p_0(\mathbf{x}), \ldots, p_{C-1}(\mathbf{x})) = \sigma(f_0(\mathbf{x}), \ldots, f_{C-1}(\mathbf{x}))$. As the underlying function $f$ is stochastic in Bayesian modeling, we can also define a posterior density $\mathrm{p}(\mathbf{p}(\mathbf{x})|\mathcal{D})$. This density informs where the model is certain or not about the class probabilities of a certain point. This uncertainty is a natural target for classification as it squashes the latent function's uncertainty when applicable, and it is purely epistemic, as it can be entirely reduced by observing more data. Hence we propose to acquire batches of points that maximize its joint differential entropy:

$$a \left( \{\mathbf{x}_b\}_{b=1}^B ; \mathrm{p}(f|\mathcal{D}) \right) = \mathbb{H} \left( \{\mathbf{p}(\mathbf{x}_b)\}_{b=1}^B \right).$$

In order to make this tractable, we make two simplifying assumptions: first, that the class probabilities are Gaussian-distributed, and second, that each output is independent. It is then straightforward to show that this leads to the acquisition function (up to constant terms)

$$a_{BBB} \left( \{\mathbf{x}_b\}_{b=1}^B ; \mathrm{p}(f|\mathcal{D}) \right) = \sum_{c=0}^{C-1} \log \det(C_p^c),$$

where $C_p^c$ is the $B \times B$ covariance matrix of the probabilities for the $c$-th class, i.e.,

$$\left( C_p^c \right)_{i,j} = \mathbb{C} \left( p_c(\mathbf{x}_i), p_c(\mathbf{x}_j) \right),$$

where $\mathbb{C}(\cdot, \cdot)$ denotes covariance. In Appendix A, we show that in binary GP classification, our acquisition function has can be written in terms of Gaussian integrals that have efficient approximations. For more general models, we adopt a sampling-based approach, using Ledoit-Wolf shrinkage [9] to ensure a well-conditioned covariance matrix (see App. B).

When we are free to choose the batch of points in $\mathcal{X}$ arbitrarily, we can simply jointly maximize the locations of $\{\mathbf{x}_b\}_{b=1}^B$ using efficient multi-start optimizers provided in standard Bayesian optimization packages [e.g., 14, 1]. However, it is more common that we will have a pool of unlabeled points to choose from, $\mathcal{D}_{\mathrm{pool}}$. In this case, the problem can be reformulated as maximizing the probability assigned to the subset of points $\{\mathbf{x}_i\}_{i=1}^B$ by the determinantal point process

$$\mathrm{P} \left( \{\mathbf{x}_i\}_{i=1}^B \right) \propto \log \det \mathbb{C} \left( \{\sigma(f(\mathbf{x}_i))\}_{i=1}^B \right).$$

While joint maximization of this probability is NP-hard, efficient greedy approaches exist which operate in $\mathcal{O} \left( |\mathcal{D}_{\mathrm{pool}}| \times B^2 \right)$ time [2].

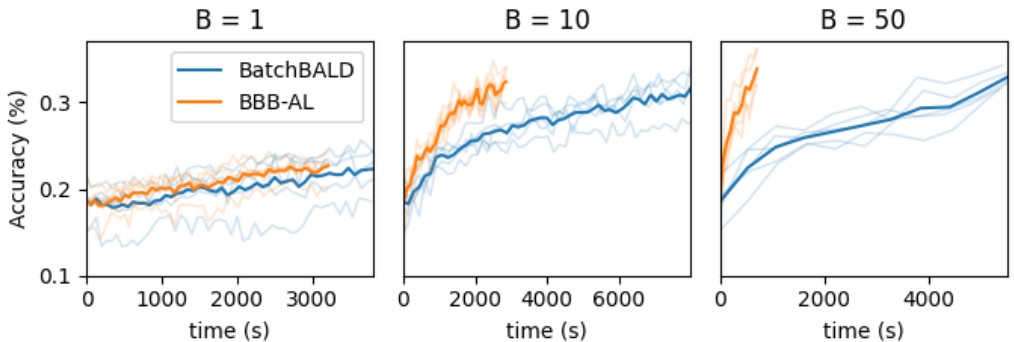

Figure 2: Accuracy on CIFAR-10 for BBB-AL and BatchBALD acquisition functions versus time for $B = 1$, $B = 10$, and $B = 50$. The darker lines show the mean performance, whereas the lighter lines show individual runs.

### 3.1 Computational complexity

In its naïve implementation, BatchBALD has a computational complexity of $\mathcal{O}(C^B \times |\mathcal{D}_{\text{pool}}|^B \times S)$, where $S$ is the number of samples from the model posterior. This high cost in $B$ is due to the need for BatchBALD to enumerate all possible combinations in $\{0, 1\}^B$. By using Monte Carlo sampling of these combinations to approximate the objective, Kirsch et al. [7] reduce this to $\mathcal{O}(BCM \times |\mathcal{D}_{\text{pool}}| \times S)$ complexity, where $M$ is the number of Monte Carlo samples. For BBB-AL, we have a time complexity of $\mathcal{O}(C \times B^2 \times |\mathcal{D}_{\text{pool}}|)$ for GP models. For more general models, this becomes $\mathcal{O}(S \times C \times |\mathcal{D}_{\text{pool}}| + C \times B^2 \times |\mathcal{D}_{\text{pool}}|)$ computational cost. For small $B$, we expect this to be better than BatchBALD, and below, we show that in practice we are significantly faster than BatchBALD regardless of batch size.

## 4 Experiments

We demonstrate our approach on CIFAR-10 [8] using the small "ResNet-8" convolutional network from Ober and Aitchison [13]. We use their fac $\rightarrow$ gi variational posterior with a learned variance (per layer) Gaussian prior, which was shown to provide a good trade-off between performance and computational complexity, using 100 inducing points, horizontal flipping and random cropping as data augmentations, and KL tempering with a factor of 0.1. We initialize the model with 50 randomly-selected points, and acquire the greater of 50 acquisitions or 500 points for batch sizes 1, 10, and 50. We repeat each experiment 5 times, and plot the accuracy as a function of time in Fig. 2. Note that we only plot the time related to acquisition, and so we exclude the time related to model training.

We observe first that our proposed approach gives marginally better results on $B = 1$, even in terms of accuracy: this may be somewhat surprising, as our approach was motivated by the batch setting. However, we believe this validates our argument that BALD can conflate epistemic and aleatoric uncertainty. For larger batch sizes, our approach is clearly significantly faster than BatchBALD, while again obtaining better outright performance.

## 5 Related work

Beyond BatchBALD [7], perhaps the most conceptually similar work to ours in terms of Bayesian active learning is given in [17]: they consider the entropy of the probabilities as we do. However, they introduce additional terms and hence propose a different objective, and most importantly they only consider single-point acquisitions. Pinsler et al. [15] provide the most relevant related work for batch active learning; however, their work assumes a true underlying posterior for a pool of data, which may not always be a realistic assumption. Our batch formulation takes inspiration from DPP-based approximations [11, 10, 12] from the related method of Bayesian optimization — where data

is collected to maximally learn about specific properties of the underlying process, rather than to reduce global uncertainty.

## 6    Conclusion & future work

In this work, we have highlighted some of the limitations of BatchBALD. To address these, we have attempted to focus our method on capturing only the epistemic uncertainty of the model. In doing so, our method also avoids the combinatorial cost of naïve BatchBALD. We have shown that our method can outperform BatchBALD in terms of both accuracy and time, allowing for bigger batches at faster runtimes.

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

## A  Expression for binary GP classification

Suppose we have a GP prior on the latent function, $f \sim \mathcal{GP}(\mu, k)$, where $\mu : \mathcal{X} \to \mathbb{R}$ is the prior mean function and $k : \mathcal{X} \times \mathcal{X} \to \mathbb{R}$ is a kernel function. To perform inference, we require a likelihood, $\mathrm{p}(y|f)$, which for binary classification is typically a Bernoulli likelihood with probabilities computed using an link function. In this case, we use the normal c.d.f. as the link function, resulting in a probit model:

$$\mathrm{P}(y = 1|f, \mathbf{x}) = \Phi(f(\mathbf{x}))$$
$$\mathrm{P}(y = 0|f, \mathbf{x}) = 1 - \Phi(f(\mathbf{x})).$$

For our acquisition function, we are thus interested in finding covariances of the form

$$\mathbb{C}(\Phi(f(\mathbf{x}_i)), \Phi(f(\mathbf{x}_j))).$$

In order to compute this quantity, we approximate the needed covariance by assuming that the posterior over latents, $\mathrm{p}(f|\mathcal{D})$, is also a Gaussian process, which is a common approximation in the GP literature [e.g., 4]. Under this assumption, we can write the posterior for two points as

$$\mathrm{p}(\mathbf{f}_{ij}|\mathcal{D}) = \mathcal{N}(\mathbf{f}_{ij}; \boldsymbol{\mu}, \boldsymbol{\Sigma}),$$

where we have defined

$$\mathbf{f}_{ij} := \begin{pmatrix} f(\mathbf{x}_i) \\ f(\mathbf{x}_j) \end{pmatrix}.$$

Recall that

$$\mathbb{C}(\Phi(f(\mathbf{x}_i)), \Phi(f(\mathbf{x}_j))) = \mathbb{E}[\Phi(f(\mathbf{x}_i))\Phi(f(\mathbf{x}_j))] - \mathbb{E}[\Phi(f(\mathbf{x}_i))]\mathbb{E}[\Phi(f(\mathbf{x}_j))].$$

By using the definition of the c.d.f. $\Phi(\cdot)$, and using the shorthand $f_i := f(\mathbf{x}_i)$, the first term in this expression can be computed as

$$\mathbb{E}[\Phi(f_i)\Phi(f_j)] = \int \Phi(f_i)\Phi(f_j)\mathcal{N}(\mathbf{f}_{ij}; \boldsymbol{\mu}, \boldsymbol{\Sigma})\,\mathrm{d}\mathbf{f}_{ij}$$

$$= \int \left( \int \mathbb{I}[y_1 \leq f_i, y_2 \leq f_j]\mathcal{N}(\mathbf{y}; \mathbf{0}, \mathbf{I}_2)\,\mathrm{d}\mathbf{y} \right) \mathcal{N}(\mathbf{f}_{ij}; \boldsymbol{\mu}, \boldsymbol{\Sigma})\,\mathrm{d}\mathbf{f}_{ij} \quad (1)$$

$$= \int \left( \int \mathbb{I}[z_1 \geq 0, z_2 \geq 0]\mathcal{N}(\mathbf{z}; \mathbf{f}_{ij}, \mathbf{I}_2)\,\mathrm{d}\mathbf{z} \right) \mathcal{N}(\mathbf{f}_{ij}; \boldsymbol{\mu}, \boldsymbol{\Sigma})\,\mathrm{d}\mathbf{f}_{ij} \quad (2)$$

$$= \int \mathbb{I}[z_1 \geq 0, z_2 \geq 0]\mathcal{N}(\mathbf{z}; \boldsymbol{\mu}, \boldsymbol{\Sigma} + \mathbf{I}_2)\,\mathrm{d}\mathbf{z} \quad (3)$$

In this derivation, we have defined $\mathbf{y} = (y_1, y_2)^\top$ and $\mathbf{z} = \mathbf{f}_{ij} - \mathbf{y}$. Eq. 1 results from the definition of the normal c.d.f., Eq. 2 comes from substituting $\mathbf{z}$ in as defined, and the final Eq. 3 is a result of marginalization. The final equation is the probability assigned to the event that $\mathbf{z} \geq \mathbf{0}$ by a normal distribution with mean $\boldsymbol{\mu}$ and covariance $\boldsymbol{\Sigma} + \mathbf{I}_2$. While this quantity does not have a closed-form expression, efficient approximations exist for the normal case [3].

We now turn to the remaining expectation, which we can derive a closed-form expression for using similar tricks:

$$
\begin{aligned}
\mathbb{E}\left[\Phi\left(f_i\right)\right] &= \int \Phi\left(f_i\right) \mathcal{N}\left(f_i; \mu_i, \sigma_{ii}^2\right) \mathrm{d}f_i \\
&= \int \left(\int \mathbb{I}\left[y \leq f_i\right] \mathcal{N}\left(y; 0, 1\right) \mathrm{d}y\right) \mathcal{N}\left(f_i; \mu_i, \sigma_{ii}^2\right) \mathrm{d}f_i \\
&= \int \left(\int \mathbb{I}\left[z \geq 0\right] \mathcal{N}\left(z; f_i, 1\right) \mathrm{d}z\right) \mathcal{N}\left(f_i; \mu_i, \sigma_{ii}^2\right) \mathrm{d}f_i \\
&= \int \mathbb{I}\left[z \geq 0\right] \mathcal{N}\left(z; \mu_i, \sigma_{ii}^2 + 1\right) \mathrm{d}z \\
&= \Phi\left(\frac{\mu_i}{\sqrt{\sigma_{ii}^2 + 1}}\right).
\end{aligned}
$$

We are now able to calculate our desired acquisition function for a set of points $\{\mathbf{x}_b\}_{b=1}^B$.

## B   Sample-based acquisition

To allow for more general models than a GP, we use a sample-based approach, in which we sample $S$ functions $f_s$, and build an estimator of the covariance matrix:

$$
\mathbb{C}\left(\sigma\left(f\left(\mathbf{x}_i\right)\right), \sigma\left(f\left(\mathbf{x}_j\right)\right)\right) \approx \frac{1}{S} \sum_{s=1}^S \left(\sigma\left(f_s\left(\mathbf{x}_i\right)\right) \sigma\left(f_s\left(\mathbf{x}_j\right)\right)\right) - \hat{\mu}_\sigma\left(\mathbf{x}_i\right) \hat{\mu}_\sigma\left(\mathbf{x}_j\right),
$$

$$
\hat{\mu}_\sigma\left(\cdot\right) = \frac{1}{S} \sum_{s=1}^S \sigma\left(f_s\left(\cdot\right)\right).
$$

Naïvely applying this estimator of the covariance matrix, however, has two issues: first, the covariance matrix will be degenerate when $B \geq S$, and second, even if it is not degenerate, the resulting covariance matrix may be numerically unstable. To address these issues, we use Ledoit-Wolf shrinkage [9], a popular covariance estimator in mathematical finance which will give us a full-rank covariance matrix. As we do not wish to evaluate the Ledoit-Wolf shrinkage factor using the full $|\mathcal{D}_{\text{pool}}| \times |\mathcal{D}_{\text{pool}}|$, we use a randomly-selected subset of our pool data to do so.

