# OpenReview forum: "Big Batch Bayesian Active Learning by Considering Predictive Probabilities"
_NeurIPS.cc/2024/Workshop/BDU — NeurIPS BDU Workshop 2024 Poster_

### Official Review · Reviewer_YvSg · 2024-09-26
**Review of Big Batch BAL**

**Rating:** 6
**Confidence:** 4

**Review:**

**Summary:**
The paper proposes an extension to batchBALD to perform batch active learning. Two critics of batchBALD were discussed: (1) computational cost, (2) the tendency to choose similar points in the batch (illustrated with a one-dimensional example). The authors proposed a new acquisition function in the form of entropy of predicted probabilities and show that it overcomes both issues.

**Strength:**
- The proposed acquisition function takes a simple and elegant form and is intuitive to understand.
- The simulation results show a clear advantage of the model over batchBALD.

**Weakness:**
- While intuitive, the proposed acquisition function strikes me as arbitrary. For example, why entropy of p rather than f? Unlike BALD, it does not arise from any principled / interpretable quantities of interest. I wonder if the authors can expand on why they decide to use this acquisition function.
- It is also unclear to me why the Gaussian assumption for class probabilities (which lives on a simplex), other than the computational appeal. I assume there is a more rigorous justification for this assumption based on asymptotic normality? But right now the lack of justification seems insufficient to me.
- The use of Ledoit-Wolf shrinkage to stabilize the covariance in high dimensions also seems to be an arbitrary choice of convenience and it is unclear how the results depend on the specific heuristic steps.

**Overall:**
In general, I like the idea and the performance. All the weaknesses I see come from the lack of more rigorous justification of the choices made along the way. The simulation analysis also seems thin to me in terms of the scenarios considered and the number of experiments performed, but I understand that for a workshop paper.

---

### Official Review · Reviewer_Chq7 · 2024-09-27
**review on paper35**

**Rating:** 9
**Confidence:** 4

**Review:**

The paper addresses an interesting and relevant problem in the field of active learning, specifically within the Bayesian framework. The authors propose an acquisition function that differentiates between epistemic and aleatoric uncertainty, which is a significant step towards optimizing batch active learning processes. The approach to focus solely on epistemic uncertainty to streamline computation and potentially enhance performance in active learning scenarios is both novel and promising.

(1) Clarity and Precision: The manuscript is well-written with clear objectives and a straightforward methodology.

(2) Theoretical Contribution: The proposal to separate epistemic and aleatoric uncertainty in the acquisition function is theoretically appealing. This could significantly reduce computational overhead and refine the selection process in active learning.

---

### Decision · Program_Chairs · 2024-10-09

Accept (Poster)